# Research Progress on Catalytic Combustion of Volatile Organic Compounds in Industrial Waste Gas

Kai Li [1] and Xiaoqing Luo [2,3,*]

1   School of Naval Architecture and Maritime, Guangdong Ocean University, Zhanjiang 524055, China
2   College of Ocean and Meteorology, Guangdong Ocean University, Zhanjiang 524088, China
3   South China Sea Institute of Marine Meteorology, Guangdong Ocean University, Zhanjiang 524088, China
*   Correspondence: gdoulxq@163.com

**Abstract:** Volatile organic compounds (VOCs) emitted from industrial processes have high stability, low activity, and toxicity which cause continuous harm to human health and the atmospheric environment. Catalytic combustion has the advantages of low energy consumption and low cost and is expected to be one of the most effective methods to remove VOCs. At present, the selection of low cost, high activity, and durability catalysts are still a difficult problem. Industrial emissions of VOCs contain a certain amount of aromatic hydrocarbons; these substances are highly toxic substances, and, once inhaled by the human body, will cause serious harm to health. In this paper, the principle, advantages, and disadvantages of VOCs processing technology are analyzed in detail, and the catalytic combustion of aromatic hydrocarbons in VOCs is reviewed, including catalyst, reaction conditions, catalyst selection, inactivation reasons, and structure use. In addition, the deactivation effects of chlorine and sulfur on catalysts during the catalytic combustion of VOCs are discussed in detail. Finally, on the basis of literature research, the prospect of catalytic combustion of VOCs is presented, which provides influential information for further research on VOCs processing technology.

**Keywords:** volatile organic compounds; treatment techniques; catalytic combustion catalyst; inactivation; regeneration

## 1. Introduction

Volatile organic compounds (VOCs) refer to the general term for various organic compounds with the lowest standard boiling point reaching 50 °C and the highest standard boiling point reaching 260 °C under normal temperature conditions. This is a very common and polluting organic substance. VOCs are highly toxic, polluting, and irritating and will have a serious impact on the atmospheric environment, including various types of organic substances, such as oxygen-containing organic compounds and nitrogen-containing, halogenated hydrocarbons, non-methane hydrocarbon organic compounds. These organic compounds can cause damage to the human body, such as deformity, gene mutation and cancer. Studies have shown that most VOCs can easily cause acute poisoning when the concentration is too high. In mild cases, dizziness, cough, and nausea may occur, and in severe cases, coma or even life-threatening; some flammable and explosive VOCs can also cause fires and explosions [1,2]. VOCs emitted by industry are harmful to human health and the ecological environment, and the control and governance of VOCs are one of the focuses of modern environmental work.

With the continuous strengthening of air pollution control, the control of PM2.5, PM10, $SO_2$, $NO_2$, and other pollutants has achieved remarkable results, but the total amount of VOC emissions is on the rise [3,4]. As people began to pay attention to the problem of air pollution, the governance of VOCs has become a hot research topic for researchers. With the help of researchers, various treatment techniques have emerged.

Catalytic combustion, also known as flameless combustion, is widely used because of its product without secondary pollution, simple process operation, low energy consumption, and high safety. Catalytic combustion technology has become one of the most cost-effective VOC treatment technologies. The research and development of catalysts is the key content of catalytic combustion technology, which determines the quality of catalytic combustion, so it has become a research hotspot.

As show in Figure 1, the VOCs emission sources in industrial production mainly include the coal chemical industry, petrochemical industry, shipbuilding industry, industrial adhesive manufacturing, industrial washing solvent, household chemical production, etc. The VOCs of industrial emissions contain a certain amount of aromatic hydrocarbons, such substances are highly toxic substances.Therefore, the catalytic combustion of benzene and hydrocarbons by VOCs was studied in this paper. This paper analyzes the research progress of the treatment technology of VOCs produced by industrial emissions and briefly introduces the performance characteristics of various treatment processes. Then, the mainstream catalytic combustion treatment technology of VOCs is reviewed, and the activity, stability, and selectivity of the catalyst are discussed. Finally, the future development direction of VOCs catalytic combustion is proposed.

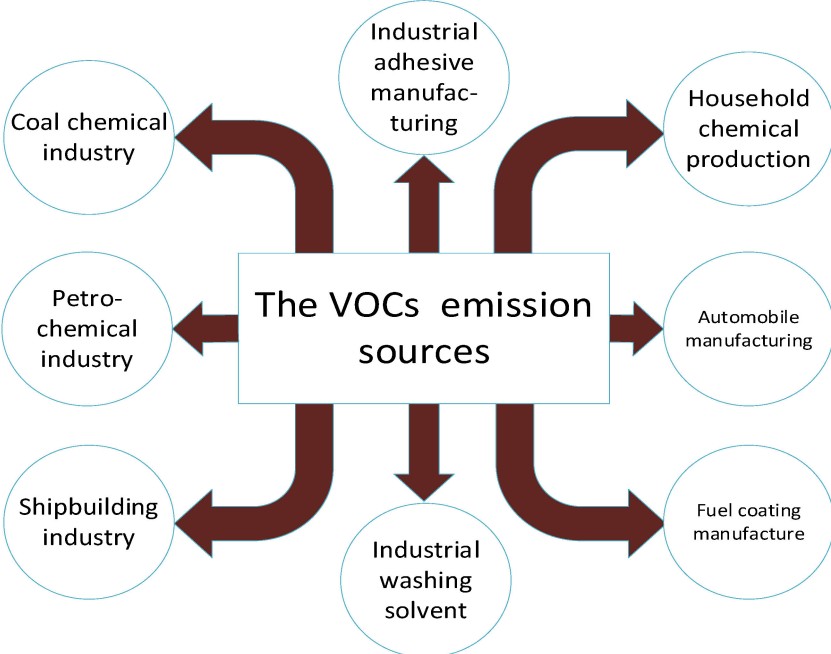

**Figure 1.** Main sources of the VOCs.

## 2. VOCs Treatment Techniques

There are many reasons for the formation of VOCs in the atmosphere, and their components are relatively complex. Therefore, the management of volatile organic compounds is difficult, and it needs to be optimized and innovated continuously. The governance of VOCs mainly includes source reduction, strict process control, and tail control treatment. Tail processing is the most important of the three links. As shown in Figure 2, the current industrial VOCs tail-end treatment technologies are divided into two types: recycling technology and destruction technology. Recycling technologies include membrane separation, absorption, adsorption, and condensation technologies. Destruction technologies include combustion, photocatalysis, biodegradation, low-temperature plasma. In the actual emission process of industrial VOCs, the gas atmosphere and other environments are more complex, and the improper selection of treatment technology will cause re-pollution and further damage the environment [5,6].

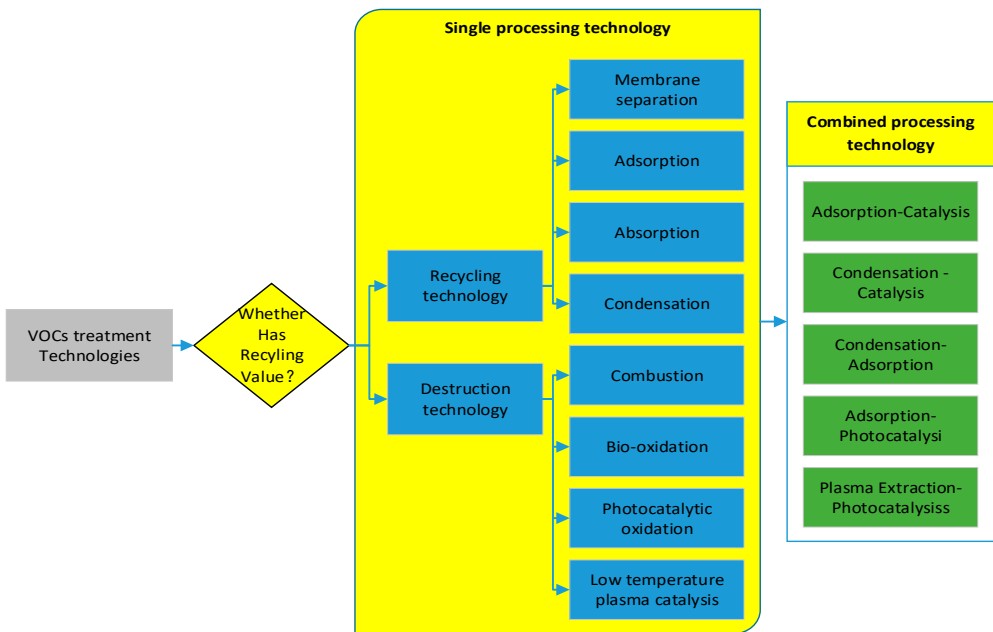

**Figure 2.** VOCs end processing technology classification architecture diagram.

## 2.1. Membrane Separation

With the help of the permeable membrane, chemical enterprises can promote the permeation of VOC waste gas components under pressure changes to achieve gas enrichment and gas separation. The entire VOCs waste gas separation and treatment application is relatively simple and convenient and can be widely used in natural gas separation and purification processes. At this stage, the widely used diaphragm material is silicone rubber. In the actual VOCs treatment, membrane separation technology can be divided into gas membrane separation method and membrane-based absorption method according to the actual situation. Membrane separation technology has many application advantages [7]. The overall operation is simple, and the treatment efficiency is high. Combining it with various processes can effectively improve the VOCs gas treatment effect and reduce costs [8].

## 2.2. Adsorption

In recovery treatment, the technology uses the adsorbent to adsorb the components with certain characteristics in the gas mixture to separate some components in the mixed gas. The commonly used adsorbents at this stage are activated carbon (granular, powdery), molecular sieve (microporous, mesoporous, and macroporous), hydrophobic silica gel, and polymer adsorption resin. Taking the activated carbon adsorbent as an example, it is characterized by a large surface area, up to 3000 m$^2$/g [9]. The raw materials are generally natural plant materials, waste rubber, and waste plastics [10].

## 2.3. Absorption

The absorption principle of VOCs is to transfer the harmful molecules in the organic waste gas to the absorbent by contacting the absorbent with the organic waste gas so as to achieve the purpose of separating the organic waste gas. According to the different working principles, this technology can be divided into chemical methods and physical methods. The physical method refers to the use of the principle of compatibility between substances and the use of absorbents to absorb harmful components in organic waste gas. The chemical method is to remove harmful molecules through the chemical reaction between organic waste gas and solvent. For solvent-insoluble waste gas, it can only be treated by chemical methods. In terms of application, physical methods are more extensive than chemical methods. The commonly used absorbents are kerosene and diesel. The absorption process requires frequent replacement of the absorbent during the action

process, which is troublesome to operate and has a high economical cost. The research and development of new absorbents and the optimal design and manufacture of absorption treatment equipment are the research directions of this method [11,12].

### 2.4. Condensation

Condensation is one of the simplest methods of exhaust gas treatment. Condensation converts volatile organic compounds into liquids at lower temperatures or higher pressures to recover VOCs [13]. The saturated vapor pressure is related to the gas type and the temperature of the applied gas. The industrial condensation process can be realized by increasing the pressure under the condition of constant temperature, and it can also be realized by decreasing the temperature under the condition of constant pressure. Using the condensation method, the exhaust gas can be purified to a high degree, but the high purification requirements are often not achieved by cooling water at room temperature. The higher the purification requirements, the lower the cooling temperature, and the pressure must be increased if necessary, which will increase the difficulty and cost of the treatment process [14]. Therefore, condensation methods are often used in conjunction with adsorption, combustion, and other purification methods to treat VOCs [15].

### 2.5. Bio-Oxidation

The ideal VOC gas treatment effect can also be achieved through the application of biological treatment technology. This technology uses the action of microorganisms to complete the transformation and degradation of VOCs gas and forms low-harm or harmless substances. At this stage, many biological treatment technologies are available. Biological treatment is used in the field of foul-smelling organic waste gas. The application of this technology can directly and effectively reduce organic molecules, improve the removal rate of VOCs gas, lower the cost, and achieve a high removal rate of waste gas that can reach 95% [16]. However, if biological treatment technology is used, it is only suitable for waste gas with low concentrations itself. Completing high-concentration waste gas treatment and recycling VOCs gas that has been treated is difficult. The application of biological treatment technology to complete VOC treatment often requires the application of biological scrubbers, biological filters, and the entire treatment. It can also be used in combination with membrane separation technology to form a complete adsorption and degradation treatment process. The reaction conditions required by biological treatment oxidation technology are relatively harsh, such as suitable pH value, VOC concentration, and molecular weight [17].

### 2.6. Photocatalytic Oxidation

The use of different light irradiation to degrade volatile organic compounds in the atmosphere can scientifically and reasonably improve the air pollution problem. The principle of photocatalytic oxidation technology is that the photocatalyst generates strong oxidizing hydroxyl radicals under direct sunlight for a long time, degrades volatile organic substances in the atmosphere, and then produces non-polluting inorganic substances through a series of chemical reactions [18]. In the process of VOCs gas treatment in chemical enterprises, through the application of photocatalytic oxidation technology, the strong decomposition of various viral organic compounds can be achieved, and various malodorous gases, including methyl sulfide, hydrogen sulfide, and trimethylamine, can be cracked. It can directly break the molecular chain of various polymer malodorous compounds and form a good effect, especially in the case of ultraviolet beam irradiation; it can further degrade and transform low molecular compounds, reduce environmental pollution, and achieve the effect of environmental protection [19].

### 2.7. Low-Temperature Plasma Catalysis

The low-temperature plasma catalysis is a method for the treatment of medium and low concentration organic waste gas. VOCs react with these active groups with

higher energy; some of them will be cracked and finally converted into substances such as carbon dioxide and water so as to achieve the purpose of purifying exhaust gas. This method can effectively remove volatile organic compounds (VOCs), inorganic substances, hydrogen sulfide, and other major pollutants and can be used to treat and purify painting workshops, ink printing, painting workshops, chemicals, medicine, rubber, food, toxic and harmful waste gas and odor generated in the production process of printing and dyeing, papermaking, and brewing. In addition to being closely related to the electrode voltage, the efficiency of this method to degrade VOCs is also affected by temperature, relative humidity, oxygen content, gas concentration, and airflow, among which the exhaust gas concentration and airflow are the main ones [20,21].

*2.8. Combustion*

In the production process of chemical enterprises, a large amount of VOC gas will be formed, which will have a serious impact on the ecological environment and the lives of residents. In order to achieve the control of VOCs gas, combustion treatment can also achieve good waste gas control effects. Specifically, combustion technology refers to the direct discharge of various waste gases into the combustion chamber so that air is introduced into the combustion chamber so that the VOCs gas can be fully and thoroughly combusted in the combustion chamber, and can be decomposed under the action of combustion to generate CO or $CO_2$ and $H_2O$. Even if the exhaust gas is burned in the combustion chamber, there will be differences in the combustion methods used, which can be divided into direct combustion and catalytic combustion [22,23].

(1) Direct combustion

The direct combustion method, also known as the thermal combustion method, is a technology in which the temperature of the exhaust gas is raised to a temperature that can be combusted by itself. This technology is only suitable for the treatment of exhaust gases containing higher concentrations of combustible gas components or higher calorific values. The combustion temperature is generally set at about 1100 °C [24]. In order to fully burn the exhaust gas, it is necessary to mix the high concentration of the exhaust gas with the air before combustion to avoid the production of harmful substances, such as dioxins.

(2) Catalytic combustion

Direct incineration is implemented at a higher reaction temperature, so its energy consumption and operating costs are higher. Catalytic combustion has attracted the great attention of researchers. Catalytic combustion has been widely used in the field of exhaust gas treatment due to its ability to convert VOCs into CO or $CO_2$ and $H_2O$ at lower temperatures. Catalytic combustion refers to the process in which VOCs are oxidized to form $CO/CO_2$ and $H_2O$ at low temperatures under the catalysis of oxidation catalysts [25]. This treatment technology has the advantages of high oxidation efficiency, low light-off temperature, low energy consumption, and less secondary pollution [26]. The key to this technology is to develop high-performance catalysts with excellent low-temperature oxidation activity, high selectivity, and strong resistance to chlorine toxicity. In addition, the use of suitable catalysts can enable catalytic combustion of VOCs to be carried out at much lower temperatures than direct combustion [27].

**3. Research Progress of Catalytic Combustion Catalysts**

*3.1. Precious Metal Catalyst*

Among the VOC catalytic combustion catalyst systems, noble metal catalysts have been used in practical VOC treatments due to their high catalytic activity and low light-off temperature. Noble metal catalysts are mainly composed of noble metals such as Pd, Pt, Au, and Ag, and their catalytic oxidation activity is easily affected by other factors, such as the addition of a second metal component. Fu et al. [28] achieved complete oxidation of toluene with Pt-Pd/MCM-41 two-component noble metal catalysts at 180 °C, which is significantly better than the one-component catalysts with the same noble metal content (Pt/MCM-

41 and Pd/MCM-41).In reference [29], the author's experimental results indicated that Pt-Pd/TiO has more adsorbed oxygen species than the Pd/TiO catalyst, which is more conducive to the transport of intermediates and adsorbed oxygen, thus speeding up the oxidation reaction rate and improving catalytic activity. The catalytic performance of a noble metal catalyst depends on its preparation method, the properties of the noble metal itself, reaction conditions, and other factors. However, the cost of noble metal catalysts is high, and sintering and poisoning are likely during the catalytic process, which makes the catalyst ineffective.

Pt-based catalysts are widely used in the catalytic oxidation of VOCs due to their high activity and high stability, especially for the catalytic oxidation of benzene, toluene, and ethylbenzene. Pt catalyst is especially suitable for the removal of aromatic compounds, and for alkanes, it has a good degradation effect on substances above pentane [30]. In addition, the active phase Pt keeps its physicochemical properties unchanged because it does not directly interact with the carrier during the impregnation process. However, the physicochemical properties of the support may affect the deposition and dispersion of Pt, thereby affecting the performance of the catalyst, including its resistance and stability to poisoning. Yang et al. loaded single-atom Pt on three-dimensional ordered mesoporous $Fe_2O_3$ for the catalytic oxidation of benzene and found that the conversion of benzene could reach 90% at a temperature of 198 °C with good hydrothermal stability. The main reason for this phenomenon is the strong interaction between Pt and the carrier $Fe_2O_3$ [31]. Peng et al. studied the effect of the size of Pt on the catalytic effect of Pt-based catalysts. Pt particles ranging from 1.3 nm to 2.5 nm were prepared and supported on $CeO_2$ to investigate the degradation effect of toluene. The results found that the reaction rate of toluene was affected by Pt. The size and size of oxygen vacancies are influenced by two factors, and the 1.8 nm $Pt/CeO_2$ has the best catalytic effect due to the balance of oxygen vacancies and Pt size [32]. Li et al. found that Pt-supported $CoO_x$ exhibited excellent catalytic performance for the degradation of toluene, which was mainly due to the strong oxygen mobility, structural defects, and abundant reactive oxygen species provided by $CoO_x$. The interaction enhanced the dispersion of Pt species and produced more $Pt_0$ [33]. The adsorption of Pt-based catalysts on the support surface will fail under humid conditions. Sedjame et al. used Pt-supported $CeO_2$-$Al_2O_3$ to degrade n-butanol and acetic acid. The study found that adding ceria to alumina reduced the surface area and changed the physicochemical properties and activity of the catalyst, which enhanced the oxidation performance of the catalyst to acetic acid but had no obvious effect on the catalytic oxidation of n-butanol, which indicated that the nature of VOCs itself had influence on the performance of the catalyst.

Compared with other noble metal catalysts (Au, Pt, Ru), Pd-based catalysts are widely used in the catalytic combustion technology of industrial VOCs due to their electronic structure and higher chemical stability during the catalytic oxidation of VOCs [34]. In general, factors affecting the catalytic activity of supported Pd-based catalysts include the particle size of Pd, the chemical state of Pd, and the type of support, acidity, and metal-support interactions. Giraudon et al. [35] found that reducing the size of Pd nanoparticles can increase the interface between Pd and the support and allow faster recovery of oxygen vacancies for better dispersion, thereby providing more available mobile oxygen. The chemical state of Pd is one of the important factors affecting the catalytic activity of Pd-based catalysts, but which valence state plays a role is still controversial. Wang et al. explored the effect of Pd valence on Pd-based catalysts in the catalytic oxidation of o-xylene at low temperatures and found that for the oxidation of o-xylene at low temperature, the activity of Pd in the metal valence state was higher than that of the Pd species in the oxidation valence state. The reducing oxide with higher activity is used as the carrier and exists independently [36]. He et al. [37] showed that in the presence of $O_2$, part of the metal Pd will be oxidized to $Pd^{2+}$, which is subsequently reduced by HCs, and pointed out that the mixture of metal Pd and $Pd_{2+}$ can also accelerate the oxidation of benzene, toluene, and ethyl acetate in the Pd/SBA-15 catalyst. Pérez et al. [38] showed that the catalytic activity

of Pd surface-supported on carbon-based materials for the combustion of m-xylene had something to do with the surface area of the support, and the mesoporous samples were considered to have the best performance compared to other forms of structures. The high activity of calcified shrimp shell-supported Pd nanoparticles for benzene degradation by Odoom et al. [39] depends on the synergistic reaction between the small Pd nanoparticles and the shrimp shell carrier and the high dispersion of Pd.

Since Au is almost considered to be chemically inert, Au-based catalysts generally have poor catalytic activity. Tends to remove benzene, toluene, ethylbenzene, etc. at higher temperatures (in the range of 190 °C to 400 °C) compared to Pd-based catalysts, Pt-based catalysts [40]. However, Au-based catalysts do not generate carbon deposits as by-products of incomplete combustion in the process of oxidizing benzene, toluene, and ethylbenzene. Metal-oxygen (M-O) bond formation is the rate-determining step in Au oxides, whereas M-O bond cleavage is slower in Au oxides compared to most other metal oxides. Therefore, most of the catalyst surfaces are in a metallic state and cannot exchange oxygen, which may negatively affect the oxidation efficiency [41]. Perez et al. [42] prepared Au/3D-ordered macroporous $LaCoO_3$ by PVA-protected reduction method for the catalytic oxidation of toluene. The strong interaction between $LaCoO_3$ may be the reason for the excellent catalytic performance of 7.63 Au/3DOM $LaCoO_3$ [43]. Carabineiro et al. [44] supported Au on different commercial oxide supports for the catalytic oxidation of ethyl acetate as well as toluene and found that Au supported on CuO and NiO yielded the best catalytic results. Precious metal catalysts are very active in the reaction and provide more oxygen for the reaction in catalytic combustion, which can effectively speed up the reaction speed and reduce the light-off temperature requirements for the reaction. It is widely used in the catalytic oxidation of polycyclic aromatic combustion.

In recent years, Ag has been widely used in the catalysis of VOCs, silver-containing catalysts, and is low toxicity, low cost, and very active in the oxidation of some VOCs. $Ag/CeO_2$ materials have high active activity, and the high dispersion of silver on $CeO_2$ enhances its oxidation compared to Ag or $CeO_2$ components alone. Compared with other metals, the price of silver is lower, and the content is higher when high temperature is not required, which can maintain higher activity and activity as well as durability [45]. Among catalysts that remove VOCs and bacteria at the same time, catalysts containing copper (Cu)/silver (Ag) have a good antibacterial effect while maintaining high activity and can be used as a potential new method to effectively prevent bacterial inactivation [46]. Ag-MnO has the highest intrinsic activity and is a good candidate for eliminating VOCs and ozone, and can be used as a catalyst for air pollution control [47]. Table 1 lists the Performance of some noble metals catalysts for the catalytic combustion of VOCs.

**Table 1.** Performance of some noble metals catalysts for the catalytic combustion of VOCs. Adapted with permission from Ref. [48]. Copyright 2019 Han Gao.

| Catalyst | VOCs | Space Velocity | Mass Fraction of Precious Metals | Temperature/°C | Ref. |
|---|---|---|---|---|---|
| Pd-Pt/SiO$_2$ | toluene | 6000 mL/gh | 0.25%Pt–0.25%Pd | 146 | [49] |
| Pt-CoAlO | toluene | 60,000 mL/gh | 3.42%Pt | 282 | [50] |
| Pd-CoAlO/Al$_2$O$_3$ | toluene | 60,000 mL/gh | 3.26%Pd | 192 | [51] |
| Pd/CeTiO$_x$ | toluene | 60,000 mL/gh | 1.5% Pd | 220 | [52] |
| Pd/ZSM-5-KIT-6 | toluene | 32,000 h$^{-1}$ | 0.48% Pd | 191 | [53] |
| Pt/ZSM-5 | toluene | 120,000 mL/gh | 2.0%Pt | 146 | [54] |
| Pt/TiO$_2$ | toluene | 30,000 mL/gh | 0.4% Pt | 137 | [55] |
| Pd/OMS-2 | toluene | 240,000 mL/gh | 0.5%Pd | 240 | [56] |
| Pt-Pd/MCM-41 | toluene | 10,000 h$^{-1}$ | 0.2% Pt–0.1% Pd | 162 | [28] |

**Table 1.** *Cont.*

| Catalyst | VOCs | Space Velocity | Mass Fraction of Precious Metals | Temperature/$^\circ$C | Ref. |
|---|---|---|---|---|---|
| Pt/FeCrAl | toluene | 10,000 h$^{-1}$ | 0.20% Pt | 160 | [57] |
| Pd-MoO$_3$/Al$_2$O$_3$ | benzene | 4800 h$^{-1}$ | 1.0%Pd | 180 | [58] |
| Pt/Nd/MCM-41 | benzene | 20,000 h$^{-1}$ | 0.2% Pt | 200 | [59] |
| Pd/Ce/shell-powder | benzene | 20,000 h$^{-1}$ | 0.2%Pd | 240 | [60] |
| Pt-Pd/Ce$_{0.75}$Zr$_{0.25}$/Al$_2$O$_3$ | benzene | 20,000 h$^{-1}$ | 0.1% Pt-Pd | 191 | [28] |
| Pd-Pt/Ce/kaolin-NaY | benzene | 20,000 h$^{-1}$ | 0.17%Pd–0.03%Pt | 180 | [61] |
| Pd-Ni/SBA-15 | benzene | 120,000 mL/gh | 0.16% Pd | 245 | [62] |
| Au/SnO$_2$ | methanol | 12,000 mL/gh | 1.7–1.8% Au | 120 | [63] |
| Au/CeO$_2$ | toluene | 60,000 mL/gh | 4.0% Au | 230 | [64] |
| Ag/CeO$_2$ | toluene | 2000 mL/gh | 2.14% | 170 | [64] |
| Au/MnOx/SiO$_2$ | toluene | 20,000 mL/gh | 0.93% Au | 234 | [65] |

### *3.2. Non-Precious Metal Catalyst*

Noble metal catalysts have many advantages, but they are expensive and single, and there are certain limitations in the treatment of organic waste gas. Many researchers focus on non-noble metal catalysts. In recent years, the exploration of non-precious metal oxide catalysts to catalyze the combustion of VOCs is one of the research hotspots in the field of environmental catalysis. V, Ce, Zr, and other non-precious metal oxides are the main components, which can be roughly divided into transition metal oxides, spinel-type, perovskite-type composite oxide catalysts and zeolite-based catalysts [66].

### 3.2.1. Transition Metal Oxide Catalyst

Transition metal oxides have been widely studied due to their low price and high stability. Yang et al. [67] loaded CuO as the active component on SBA-15, and the results showed that when the CuO loading changed from 1% to 10%, the porosity and specific surface area of the catalyst were significantly increased, and the activity was enhanced, and it was concluded that the catalytic combustion of benzene order of activity: CuO > MnO > FeO > NiO. Compared with the single-component transition metal oxide, the two-component or multi-component transition metal composite oxide has higher active components and dispersity and thus has higher catalyst activity and better stability. Ma et al. [68] prepared a series of Fe-Mn/cordierite catalysts by impregnation method to catalyze the combustion of toluene. The results showed that when the molar ratio n(Fe):n(Mn) = 4:1, the catalysts modified by $\gamma$-Al$_2$O$_3$ coating had Optimum catalytic combustion activity. Wang et al. [69] prepared MnO$_x$, CeO$_2$, and the catalyst catalyzed the combustion of chlorobenzene. The results showed that the conversion rate of chlorobenzene reached 90% at 236 $^\circ$C, which has good anti-chlorine performance. Transition metal oxidation catalysts have attracted much attention due to their wide source of raw materials, various types, various combinations, and simple preparation processes. Manganese is one of the most abundant metals on the surface of the earth. Its compounds are low-cost and non-toxic and have simple preparation processes and high stability; as such, it is considered an "environmentally friendly" catalytic material. Manganese is also a transition metal with five unpaired electron pairs at the d level, giving it a variety of oxidation states. In the diffusion-controlled oxidation process, MnO$_x$ of different oxidation states can either coexist or be gradually converted from one form to another form of oxygen. This deficiency in the ability to switch oxidation states and form other structures facilitates high oxygen mobility and oxygen storage and stimulates the catalytic properties of MnO$_x$ [70]. A variety of MnO$_x$ were synthesized by the coprecipitation method and used for catalytic oxidation of toluene. The results show that the performance of the catalyst is also directly related to the ratio of Oads/Olatt, Which varies with the ratio of Mn$^{3+}$/Mn$^{4+}$ [71]. In their experiments, Piumetti et al. [72] found that the activity of manganese catalysts during the catalytic oxidation of VOCs is Mn$_3$O$_4$ > Mn$_2$O$_3$ > Mn$_3$O$_4$ and MnO$_2$ and pointed out that the high activity

of $Mn_3O_4$ is due to the adsorption of a large amount of electrophilic oxygen on its surface. Due to its unique oxygen release/capture ability combined with $Ce^{4+}/Ce^{3+}$ pairs, $CeO_2$ has special activity and is widely used in the catalytic combustion of VOCs [73]. Compared with other $CoO_x/g$-$CN_4$ samples, $CoO_x$ loaded with 10% $CoO_x$ load of g-$C_3N_4$ showed the best catalytic activity, good stability, and good reusability [74]. Zhang et al. [75] prepared a $Co_zO_4$ catalyst by a simple precipitation method. Through experimental comparison, it was found that Co-$CO_s$ have a good complete oxidation capacity for toluene and propane, while Co-com has low reducibility, which has a negative impact on the catalytic performance of toluene oxidation. Puertolas et al. [76] observed that the surface area of the prepared sample (obtained by a hydrothermal method in the presence of various organic acids) was reduced compared with that of the reference sample (not prepared with added organic acids) and that the reference sample had better catalytic performance than the prepared sample, showing that the difference in the catalytic performance of $Co_3O_4$ was related to the specific surface area. Table 2 lists the effects of non-noble metal catalyst on the catalytic combustion of VOCs.

**Table 2.** Effects of non-noble metal catalysts on the catalytic combustion of VOCs. Adapted with with permission from Ref. [77]. Copyright 2021 Xiaotian Mu.

| Catalyst | VOCs | Space Velocity/h$^{-1}$ | Concentration/10$^{-6}$ | Conversion Rate/% | Temperature/°C | Ref. |
|---|---|---|---|---|---|---|
| $Mn_3AlO$ | Acetone | 18,000 | 500 | — | 183 | [78] |
| $\alpha$-$MnO_2$ | Acetone | 9,000 | 1000 | — | 104 | [79] |
| $CeO_2$ (Ethylene glycol combustion preparation) | naphthalene | 45,000 | 450 | 100 | 250 | [80] |
| $CeO_2$(Preparation of urea precipitationi | naphthalene | 25,000 | 100 | 100 | 175 | [81] |
| Nanometer $CeO_2$ | naphthalene | 25,000 | 100 | 100 | 200 | [82] |
| $Mn/TiO_2$ | PAHs | — | — | 95 | 395 | [83] |
| $CeO_2$-$ZrO_2$ | naphthalene | — | 200 | 100 | 327 | [84] |
| $CeO_2$–$CoO_x$-10 | toluene | 78,000 | 1000 | 90 | 248 | [85] |
| $Co_3O_4$-400(MOFs) | toluene | 21,000 | 12,000 | >90 | 259 | [86] |

### 3.2.2. Perovskite Oxide Catalysts

Perovskite oxide refers to a composite oxide with natural perovskite type formed by rare earth and transition metal oxide under certain conditions. The oxygen vacancies formed by lattice distortion in perovskite catalysts have the function of transporting and storing oxygen, and their ideal cubic crystal structure gives these catalysts the characteristics of chemical corrosion resistance and thermal stability. The general structure of perovskite catalysts is $ABO_3$, A and B represent trivalent cations, and the higher the bond energy of the metal oxide at the B position, the higher the catalyst stability. Due to the absence of an oxygen-rich state in the anionic defect structure, the perovskite structure is destroyed, resulting in deactivation, while the oxygen-rich state of the cationic defect structure makes the surface oxygen vacancies react with impurities such as chlorine, ensuring the integrity of the perovskite structure. A large number of studies have shown that the conversion rate of chlorobenzene is proportional to the content of active oxygen adsorbed on the catalyst surface, which is because the adsorbed oxygen is related to the migration rate of chlorine species, while $La_{0.8}Sr_{0.2}MnO_3$ has higher redox ability and oxygen mobility and, therefore, high stability. Huang et al. [87] studied perovskite-type metal oxide catalysts to catalyze the oxidation of VOCs, prepared La1-xSrxCoO$_3$ (x = 0, 0.2) catalysts by co-precipitation method and compared with the previously developed $LaCoO_3$ perovskite-type catalysts and found that, The catalytic activity of $La_{0.8}Sr_{0.2}CoO_3$ is much higher than that of $LaCoO_3$ type catalyst. Zhang et al. [88] used the traditional precipitation method to prepare $LaMnO_3$ and $LaB_{0.2}Mn_{0.8}O_3$, where B can be Fe, Co, Ni, and perovskite oxides and studied them as catalysts for oxidizing vinyl chloride at a temperature of 50~350 °C. The results show that the mixed $LaB_{0.2}Mn_{0.8}O_3$ sample exhibits higher activity than pure $LaMnO_3$, which is due

to the fact that the catalytic activity of perovskite oxides is related to the low-temperature reducibility of B sites and the amount of adsorbed oxygen, and the oxygen vacancies on the catalyst surface. Hu et al. [89] prepared the $MnOx/Co_3O_4$ catalyst by impregnation method using Co-ZIF-67 as a template and found that its catalytic oxidation of chlorobenzene showed excellent catalytic activity compared with commercial $Co_3O_4$, Mainly because of its unique physical and chemical properties. Due to the different activation energies of various organic compounds, the effect of different catalysts on catalyzing different organic compounds is very different. Perovskite-type complex oxides generally show good catalytic activity, but there are still many theoretical and practical problems in the current preparation research. To be resolved, such as the ratio of components, life and, other issues need further testing and research.

### 3.2.3. Spinel-Type Oxide Catalyst

Spinel-type composite metal oxide is an important structural type of combustion catalyst with the general formula $AB_2O_4$, which has good deep catalytic oxidation activity. Morales et al. [90] prepared Mn-Cu series catalysts by co-precipitation method and studied the performance of the catalysts in the catalytic oxidation of ethanol. The experimental results show that the co-precipitation method can synthesize Mn-Cu catalysts with excellent catalytic performance, which is beneficial to the improvement of the activity. Hosseini et al. [91] prepared $MCr_2O_4$ type catalysts by sol-gel method. The weak binding ability of $Cr^{6+}$-O-$Cr^{6+}$ sites to oxygen makes $ZnCr_2O_4$ and $CuCr_2O_4$ have excellent stability. Among them, $ZnCo_2O_4$ exhibits better properties than the other two composite oxides and is a promising catalyst. The catalytic performance of this type can be further improved by mixing other elements or using different precursors. Jiratova et al. [92] mixed different amounts of potassium (mass fraction between 0 and 3%) into the Co-Mn-Al catalyst to examine the effect of potassium on the deep catalysis of toluene by the catalyst. The experimental results show that the addition of potassium changes the performance of the Co-Mn-Al catalyst. When the addition amount is 1% by mass, the catalytic efficiency of toluene is the highest, and no other products are produced except water and carbon dioxide. Siham et al. [93] used low-cost biopolymer alginate as a precursor to prepare three nano-scale oxides of $Mn_3O_4$, $CuO$, $Cu_{1.5}Mn_{1.5}O_4$ with different atomic ratios of copper and manganese. His research found that the spinel-type $Cu_{1.5}Mn_{1.5}O_4$ crystal has the best activity for p-toluene, and the complete oxidation of p-toluene was achieved at 240 °C. Co spinel showed good catalytic activity, especially in the catalytic oxidation of VOCs containing benzene. Chen et al. [94] studied the method of using solar energy to recover $ACo_2O_4$ (A = Ni, Cu, Fe, Mn) spinel to remove VOCs and synthesized $ACo_2O_4$ by co-precipitation method, which could provide enough heat energy for catalytic degradation of toluene. As shown in Figure 3, the results show that the photothermal catalytic properties of $ACo_2O_4$ catalyst are $NiCo_2O_4 > CuCo_2O_4 > FeCo_2O_4 > MnCo_2O_4$. Compared with conventional catalysts, spinel-type composite oxide catalysts have excellent catalytic reaction performance and have certain application prospects. In the future, we should focus on optimizing active components, improving preparation technology, and improving catalyst dispersion and specific surface area.

### 3.2.4. Zeolite Supported Catalyst

Zeolite-supported catalysts are usually composed of catalytic active components and zeolite support. When the catalytic active components are made into a supported type, their dispersion and catalytic activity are improved, while zeolite support can provide an effective surface and appropriate pore structure, reduce the agglomeration of active components, and enhance the mechanical strength of the catalyst. In particular, zeolite molecular sieve also contains more acid sites, and has a certain catalytic activity. Due to thier adsorption and catalytic functions, zeolite-supported catalysts have been widely used in the adsorption-catalytic oxidation co-treatment of various VOCs molecular pollutants. Yang et al. [95] studied the catalytic oxidation of butane by Pt/ZAM-5 using a micro-scale

burner and found that when Pt/ZAM-5 oxidized butane, the products contained a small amount of CO, $H_2$, ethylene, and ethane, which confirmed that the oxidation of alkanes was carried out in steps. The characteristics of zeolite support have a great influence on the dispersion of active components of the catalyst. Avascues et al. [96] studied the catalytic oxidation characteristics of Pt loaded with different zeolite supports for n-hexane. The test results show that, due to the larger specific surface area and pore volume, the Pt particle size on the surface of Y zeolite is smaller, and the dispersion is more uniform. Compared with Pt/ZSM-5, Pt/Y has better catalytic activity. Huang et al. [97] found that the good adsorption performance of HZSM-5 and the REDOX performance of $MnO_x$ endowed $MnO_x$/HZSM-5 with excellent catalytic oxidation performance. Peng et Al. [98] prepared supported catalysts $MnO_x$/H-Beta and $MnO_x$/K-Beta using Beta zeolite with high Al content as the support. Their study indicated that the adsorption of oxygen on the surface of $MnO_x$ had a greater impact on the catalytic activity of the catalyst, and the $MnO_x$/H-Beta with a higher adsorption content of oxygen had a higher catalytic activity for toluene. Yosefi et al. [99] prepared a highly dispersed supported catalyst Cu-$CeO_2$/CLT by an ultrasound-chemical method. They also confirmed that the surface acidity of CLT zeolite had a great influence on the oxidation and decomposition of toluene by the catalyst, and the synergistic effect of CuO-$CeO_2$ could improve the catalytic activity of the supported catalyst for toluene. $CeO_2$ is a kind of rare earth metal oxide with good oxygen storage and release. It can not only promote the dispersion of CuO and MnO on the support but also improve the oxygen storage capacity of the catalyst so as to provide enough reactive oxygen species for the catalytic oxidation reaction. In recent years, different zeolite-supported catalysts have been widely used in the catalytic oxidation of aromatic hydrocarbons. The influence of support structure, surface physical and chemical properties, types of active components, particle size, and catalyst preparation methods on the catalyst activity and the catalytic oxidation mechanism of different aromatic hydrocarbons supported catalysts have been further elucidated. Table 3 summarizes the performance of the oxidation and pyrolysis of various aromatic hydrocarbons with different zeolite-supported catalysts. As shown in Figure 4, $ZnCoO_x$ catalysts prepared by in-situ cracking of ZnCo bimetallic zeolite imidazole salt were reasonably designed on the basis of a metal ion doping strategy. The derived $Zn_{0.05}CoO_x$ with appropriate Zn doping (Zn/Co molar ratio 0.05) has good catalytic activity to eliminate the persistence of different volatile organic compounds (VOCs).

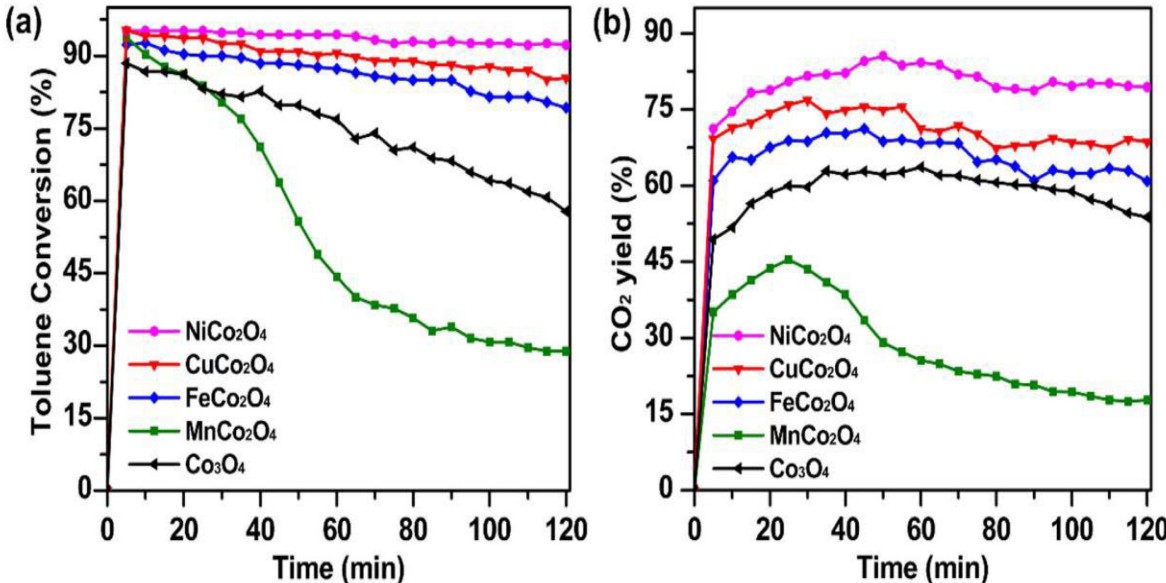

**Figure 3.** Toluene conversion (**a**) and $CO_2$ yield (**b**) over the catalysts under the irradiation (simulated sunlight, 400 mW/$cm^2$). Reproduced with permission from Ref. [94]. Copyright 2019 Xi Chen.

**Table 3.** Comparison of catalytic oxidation performance of different zeolite-supported catalysts for aromatic hydrocarbon VOCs. Adapted with permission from Ref. [100]. Copyright 2018 Aihu Feng.

| Catalyst | VOCs Concentration/ppm | VOCs | Flow Rate/(mL·min$^{-1}$) | GHSV/h | Conversion Temperature/°C | Ref. |
|---|---|---|---|---|---|---|
| 4.7%Ru | 40 | toluene | 100 | 80,000 | 251 (90%) | [101] |
| 4.5%Ru | 40 | toluene | 100 | 80,000 | 243 (90%) | [95] |
| 4.7%Ru | 40 | o-xylene | 100 | 80,000 | 257 (90%) | [95] |
| 4.5%Ru | 40 | o-xylene | 100 | 80,000 | 243 (90%) | [95] |
| 4.7%Ru | 40 | TMB | 100 | 80,000 | 278 (90%) | [95] |
| 6.7%Cu | 1000 | toluene | — | 15,000 | 293 (90%) | [102] |
| 9.55%Mn | 1000 | toluene | 100 | 60,000 | 285 (90%) | [99] |
| 10%Mn | 1000 | benzene | 125 | 20,000 | 315 (90%) | [103] |
| 10%MnCe | 1000 | benzene | 125 | 20,000 | 235 (90%) | [104] |
| 0.6% Pd-6% La | 1000 | benzene | 250 | 20,000 | 210 (90%) | [105] |

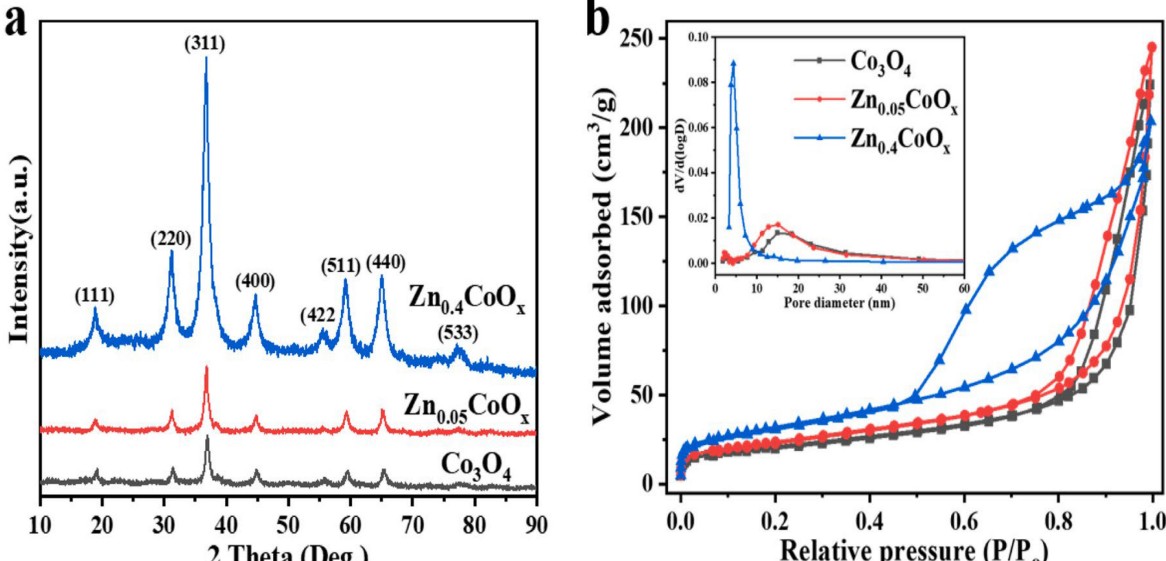

**Figure 4.** (**a**) XRD patterns and (**b**) nitrogen adsorption-desorption isotherms and the corresponding pore size distributions (inset) of $Co_3O_4$, $Zn_{0.05}CoO_x$, and $Zn_{0.4}CoO_x$. Reproduced with permission from Ref. [106]. Copyright 2022 Yunlong Guo.

### 3.3. The Use of Structured Catalysts

The important factor in the catalytic combustion efficiency of VOCs is the reasonable design of the catalyst. The traditional core-shell structure, channel encapsulation structure, two-dimensional interface-induced structure, and defect-derived structure catalysts often have some problems, such as uneven dispersion, high-temperature sintering, active and component poisoning, which affect the final performance of the catalyst. Closed structure catalyst can solve the above problems well because the active phase can be fixed in a limited space, improving its dispersion and stability. However, the existing studies on the catalytic oxidation of VOCs with limited structure catalysts are limited to some relatively simple VOC molecules. At present, there are few reports on the catalytic oxidation of VOCs containing chlorine, nitrogen, sulfur, and other heteroatoms. Future research should focus on the development of closed-structure catalysts for various VOC molecules [107]. The preparation of efficient, inexpensive catalytic combustion catalysts is the core of catalytic combustion technology. In catalytic combustion technology, catalyst performance plays a crucial role in catalytic combustion efficiency and operating cost.

## 4. Catalyst Deactivation and Inhibition

After a period of use, the activity of the catalyst gradually decreases or disappears, which is called deactivation of the catalyst. Catalyst deactivation is a core factor that directly affects the operating time of the catalyst. The methods of catalyst deactivation mainly include coke, chlorine, sulfur poisoning, water vapor, and thermal sintering. This paper discusses the problem of catalyst deactivation caused by chlorine and sulfur.

### 4.1. Chlorine Poisoning

In order to improve the anti-chlorine poisoning performance of catalysts, extensive research has been carried out on how to rapidly remove CL species adsorbed on the surface of catalysts. The optimization of the catalyst formulation is an effective means to improve the anti-chlorine poisoning performance of the catalyst [108,109]. Li et al. [110] prepared $MnO_x$-$TiO_2$ and $SnO_x$-$MnOz$-TiO catalysts by precipitation method and tested their catalytic oxidation performance of p-chlorobenzene. The experimental results show that the doping of Sn can significantly improve the stability of the Mn-Ti catalyst for the oxidation of chlorobenzene. This is because the introduction of Sn can reduce the energy required for the desorption of Cl species, which is beneficial to the removal of C1 on the catalyst surface and inhibits metal chloride. The formation of oxides, thereby improving the anti-chlorine poisoning performance of the catalyst. Ran et al. [111] prepared $Ru/TiO_z$ catalyst by impregnation method and tested its catalytic oxidation performance for Dichloromethane. The results show that the $RuTiO_2$ catalyst has good resistance and stability to chlorine poisoning, which may be related to the easy removal of the c1 species generated during the decomposition of Dichloromethane. They proposed that during the catalytic oxidation of CVOCs, the Cl species adsorbed on the catalyst surface would be transferred from $TiO_2$ to $RuO_2$ and subsequently detached as Cl or HCl.

In addition, the anti-chlorine poisoning performance of the catalyst can also be improved by adjusting the gas composition. The addition of HO in the reaction atmosphere can promote the removal of Cl species on the catalyst surface in the form of HCl, while the presence of $O_2$ is conducive to the Deacon reaction, which can promote the conversion of adsorbed Cl species into Cl. Cen et al. [112] found that when using CeO to catalyze the oxidation of CVOCs, increasing the $O_zCl$ ratio in the reactant feed can promote the removal of C1 species in the form of $C_{12}$ and effectively improve the catalyst stability. Aranzabal et al. [113] carried out experiments using H-type zeolite to catalyze the oxidation of TCE and found that the catalyst would be seriously deactivated under dry conditions, while the catalyst stability was improved after adding HO, and the surface coke and C1 content were lower. The reaction temperature is also a key factor affecting the stability of the catalyst for the catalytic oxidation of CVOCs [114]. Surface oxygen species that are inactive at low temperatures will become active when the temperature increases, which is beneficial to the continuous removal of Cl species adsorbed on the catalyst surface during the reaction process, thereby inhibiting the deactivation of the catalyst to a certain degree.

The choice of catalyst is particularly critical in the catalytic combustion of CVOCs. Precious metal catalysts are prone to chlorine poisoning. Even though the catalytic chemical activity of metal oxidation reactants is lower than that of precious metals, they can resist deactivation caused by chlorine poisoning. Composite catalysts produced by supporting them on molecular sieve intermediate carriers often have excellent CVOCs catalytic function. In addition, adding water to the reaction system can promote the chlorine substances on the surface of the catalyst at full speed and the removal of intermediate products in a certain influence environment. The addition of other hydrogen-rich additives may also reduce the chlorine attack on the catalyst by regulating the HC1 ratio of the catalytic system [115,116].

### 4.2. Sulfur Poisoning

Sulfur poisoning refers to the adsorption of $SO_z$ or $SO_3$ on the chemically active components of the catalyst and even reacts with the chemically active components of the catalyst to automatically form stable and chemically inactive sulfites and sulfates. The

sulfur-containing volatile reactive organic chemical reaction substances will form SO during the catalytic combustion process, and at the same time, there will be $SO_2$ and O and the chemically active components of the catalyst, and the chemically active oxygen species will react to automatically form SO. The researchers found the following three methods to improve the catalyst's resistance to sulfur poisoning in the catalytic combustion of VOCs.

(1) Select catalyst chemical active ingredients

The effective density of the outermost electron cloud of the chemically active components of different catalysts is different, which makes their anti-sulfide poisoning function different. Gelin et al. [117] have analyzed and studied that in the catalytic reaction without sulfur, the catalytic chemical activity of the precious metal Pd chemical active component exceeds that of the precious metal Pt chemical active component. In the sulfur-containing catalytic reaction, the chemical activity of the precious metal Pt exceeds the chemical activity of the precious metal Pd. In their experimental research, H.Ohtsuka et al. [118] found that when the precious metal is mixed with other metals, the chemically active components of the alloy have a very strong anti-SO poisoning function. The Ir/ZrO catalyst has an excellent anti-$SO_z$ poisoning function.

(2) Add auxiliary chemicals

After years of research, it has been found that adding auxiliary chemical reagents can improve the anti-sulfur poisoning function of the catalyst. Foreign experts and scholars such as Ferrandon added copper and magnesium co-chemical reagents to the palladium-based catalyst. The final analysis results of the analysis and research showed that the palladium-based catalyst with the addition of magnesium and copper has excellent anti-sulfur poisoning function. Chong et al. [119] added boron, nails, and palladium to the palladium-based catalyst. The final results of the analysis and research showed that the palladium-based catalyst with the addition of nails has an excellent anti-HS poisoning function. Fullerton et al. [120] modified the platinum-based catalyst by using the $CO_2$ reaction catalyst. The analysis and research found that the modification of the $CO_2$ reaction catalyst increased the chemical activity of the platinum-based catalyst for catalyzing the combustion of methane and, at the same time, improved the sulfur resistance of the platinum-based catalyst poisoning function.

(3) Adjust and control the acidity and alkalinity of the catalyst intermediate action carrier

Not only can the mode of adding auxiliary chemical reagents be used to improve the anti-sulfur poisoning function of the catalyst, but also the mode of adjusting and controlling the acidity and alkalinity of the intermediate-acting carrier of the catalyst can be used to improve the anti-sulfur poisoning function of the catalyst. Over the years, researchers have found through analysis and research that sulfur species adsorbs on the chemically active components of the catalyst and will obtain electrons from the chemically active components of the catalyst. Increasing the acidity of the catalyst's intermediate carrier will easily lead to the catalyst being loaded on the intermediate carrier. In the chemically active components of the catalyst, the electrons are converted to the intermediate action carrier, reducing the effective density of the electron cloud of the chemically active components of the catalyst, thereby curbing the adsorption of sulfur species on the chemically active components. Although the mode of increasing the acidity of the catalyst intermediate carrier helps to improve the chemical activity and anti-$SO_2$ poisoning function of the catalyst for catalytic combustion of VOCs polluted exhaust gas, the increase of the acidity of the intermediate carrier will increase the carbon deposition of the catalyst at the same time.

## 5. Methods to Improve Catalyst Stability

### 5.1. Monitoring Integrated Management

The effective temperature of catalytic combustion has a great influence on catalyst stability. Vu et al. [121] found that during the catalytic oxidation reaction of chlorobenzene, the $MnCuO_x/TiOz$ catalyst would be deactivated to varying degrees at 300 °C, and the

surface layer tested the automatic formation of chloride MnCuO$_x$. When the effective temperature of the reaction was increased to 350 °C, ClwTiO$_z$ can effectively prevent the deactivation of the catalyst. Although the chlorine species has not been completely eliminated from the catalyst, the regeneration of the catalyst under the high-temperature action condition restores the initial chemical activity of the reaction system. Guisnet et al. [122] found that in the process of catalytic oxidation of o-xylene, the carbon deposition on the surface of the Pd/HFAU catalyst increased with the increase of the effective temperature at low temperature, and after reaching the maximum effective value at 230 °C, it decreased as the effective temperature increases and lasted until it was completely removed at 330 °C. To sum up, high temperature helps to curb chlorine poisoning and carbon deposition of the catalyst; however, increasing the effective high temperature will lead to an increase in energy consumption, and at the same time, when the catalyst is exposed to high temperature for a long time, the thermal effect may cause Changes in catalyst structure, existing morphology and physicochemical characteristics, which in turn lead to catalyst deactivation. Water vapor is also a main factor affecting catalyst stability. Aranzabal et al. [123] proposed that in the catalytic combustion of CVOCs, water can compete with CVOCs on the catalyst for adsorption and can also promote the removal of helium bonded to molecular hydroxyl groups. In the catalytic combustion process of chlorobenzene, the addition of water can efficiently remove chlorine species and carbonaceous intermediates on the surface of Mn and Ce$_x$OHZSM-5, thus fully ensuring the stability of the catalyst [124].

*5.2. Modification of Catalysts*

The small circular pore structure, acidity, and redox function of the catalyst itself are the core factors that determine its stability, so the stability of the catalyst can be improved by a modified mode. For example, in the combustion process of CVOCs, the strong adsorption of HCl or Cl can easily lead to catalyst deactivation. According to deRivas et al. [125] analysis and research, chlorination can a cause partial surface collapse of the Ce/Zr composite oxidation reactant and damage its pore structure; but at the same time, the chemisorption of chlorine on the catalyst surface can be regarded as a brand new acid site, and the generation of Cl can enhance the oxygen transition ratio, so that the catalyst can use molecular lattice oxygen relatively easily, and improve its redox function. The loading of highly chemically active metals can improve the oxidation reaction reduction ability of the catalyst and effectively curb the loss of carbon deposits [126]. The comprehensive desiliconization treatment of MOR molecular sieve can improve the coking resistance at the same time of preservation of a certain acidity, and the comprehensive treatment of dealumination will cause the catalyst acidity to be drastically reduced, thereby reducing its chemical activity [127].

In the analysis and research of heterogeneous catalysis, the catalysts modified by alkali metal inclusions often have excellent chemical activity and stability. The inclusion of alkali metals into noble metal catalysts can promote the activation of surface molecular hydroxyl groups and the distribution of chemically active species and improve the chemical activity of formaldehyde catalytic oxidation reaction [128]. TheiInclusion of alkali metals into perovskite catalysts can promote the generation of oxygen vacancies, increase the effective concentration of oxygen species in the surface layer, and promote the dramatic reduction of the effective temperature of the catalytic combustion of soot [129]. Inclusion of alkali metals into metal oxidation reactants can weaken the Me-O bond, promote the generation and transformation of chemically active oxygen, and improve the catalytic combustion chemical activity of carbon black [130]. The addition of alkali metal to the hydrogen-type molecular sieve catalyst can adjust the effective density and functional strength of Bronsted acid on the surface of the catalyst, which also can improve the stability of the catalyst in the methane dehydroaromatization reaction [131].

### 6. Summary and Outlook

In this paper, the VOC gas emitted by industry is taken as the object, and its treatment technology is analyzed. A detailed discussion on the catalytic combustion method of the mainstream treatment technology of VOCs is carried out, and the metal catalysts are described in terms of catalyst types, active components, catalytic properties and applications, and the improvement direction of related catalysts is proposed. With the deepening of industrialization and the improvement of people's living standards, the environmental problems caused by VOCs have received more and more attention. After decades of efforts, there have been a variety of effective treatment methods for VOCs. While great progress has been made, there are still some problems to be studied.

(1)  In the catalytic combustion treatment technology, the particle size of the precious metals of the existing catalysts is still mainly nano-scale, and the loading amount is between 0.5 and 10 wt%, which leads to the high cost of some catalysts and low catalytic performance. Therefore, the preparation of low-loaded noble metal particle-supported catalysts for efficient catalytic combustion of VOCs is still a difficulty of environmental catalysis.

(2)  Compared with single-component active materials, the catalytic performance of two-component or even multi-component active materials has been significantly improved in various aspects, but the synergistic mechanism between the components is still unclear. Studying their mutual influence on particle size and dispersion degree of each other is conducive to synthesizing the most effective catalyst products under the premise of the least input.

(3)  VOCs emitted by industry are important compounds causing air pollution. With the continuous progress of the global pollution prevention and control battle, enterprises are facing unprecedented challenges to deepen the treatment of VOCs waste gas, especially for the complex composition of VOCs waste gas, difficult treatment, a single treatment plan is difficult to reach stable standards, so it is necessary to further optimize and reform the waste gas treatment technology. In the industrial field, VOC waste gas is generally treated by incineration, but transfer treatment is usually used for VOCs with low concentrations. When enterprises choose the specific treatment plan, they should determine it according to the composition, physical and chemical characteristics of the waste gas, combined with the on-site working environment conditions of enterprises (including temperature, humidity, acidity, dust, and other factors) and the local emission standards. In industrial production, enterprises can choose the joint scheme according to their own conditions and environmental needs and give play to the advantages of various methods to achieve the optimal governance effect.

(4)  With the continuous advancement of urbanization in various regions, the construction of urban infrastructure has been developed rapidly, and the impact of exhaust emissions of industrial machinery on the atmospheric environment has been paid more and more attention. The number of industrial machinery in the world is increasing year by year. Due to the huge volume of construction machinery, the complex construction environment, and the lack of a mature emission source test system, there are many deficiencies in the cognition of VOCs component characteristics of construction machinery exhaust emissions. Future research should vigorously strengthen the research on VOCs emitted by industrial machinery.

**Author Contributions:** Conceptualization, K.L.; Data curation K.L. and X.L.; Supervision, X.L.; Writing—review and editing, K.L. All authors have read and agreed to the published version of the manuscript.

**Funding:** This study was supported by Zhanjiang City Science and Technology Development Special Fund Competitive Allocation Project (NO.2021A05034) and Ph.D. project initiated by Guangdong Ocean University (060302032109).

**Institutional Review Board Statement:** Not applicable.

**Informed Consent Statement:** Not applicable.

**Data Availability Statement:** Data sharing not applicable.

**Conflicts of Interest:** The authors declare no conflict of interest.

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
