# Peer review of "Research Progress on Catalytic Combustion of Volatile Organic Compounds in Industrial Waste Gas"

_catalysts, doi:10.3390/catal13020268_

Round 1
Reviewer 1 Report
In general, the review is of interest and could help readers to introduce themselves to the subject of combustion of VOCs.
However, to be accepted, it is suggested to increase the discussion and add new citations/references related to the following points:
1- It is recommended to add citations in section 2.
2- In the section 3.1 it is recommended to broaden the discussion regarding the use of catalysts that contain gold or silver as active phases. Some recent works or recent reviews have not been considered or cited. Mainly , the catalysts that contain silver, are of low toxicity, low cost and very active for the oxidation of some VOCs.
3- Why is the discussion only limited to the use of transition metal oxide catalysts in the combustion of aromatics and polyaromatics? There is abundant literature related to the oxidation of other VOCs. If the authors decided to report only some family of compounds, it is suggested that they leave it stipulated in the Introduction.
4- Cobalt oxides (CoOx) present a high catalytic activity for oxidation/combustion reactions of VOCs. A discussion related to its use should be included in this review.
5- Could the authors include some small discussion related to mechanistic aspects?
6- It is suggested to add a small section related to the use of structured catalysts.
7- The authors must make a deep review of punctuation (missing spaces) and some editing aspects (missing subscripts, some sentences do not start with capital letters, etc.)
8- The format and text of the references should be reviewed.
Reviewer 2 Report
This review deals about the research on progress on catalytic combustion of volatile organic compounds (VOCs) in industrial waste gas. It is a topic of interest to the researchers in the related areas.
The article is difficult to read because is not well organized. The aims are ambiguous and they are not in line with the article content. Not enough references are seriously analyzed to make this kind of contribution. Literature survey must be extended in several specific points (e. g. membrane separation, absorption, adsorption, condensation, oxidation, etc.). Recent publications must be added in order to give a more complete overview of the main topic. The comments about the information chosen are too simple and some important reported articles are missing.
I found this article not suitable for publication; the level is not adequate for this Journal. Therefore, I recommend the rejection.
Additional comments:
- Some figures (e. g. Figure 4) are not relevant to the general content of the work.
- Tables are not completed with the associated references.
Reviewer 3 Report
Comments to the Authors
In this manuscript authors analyses the research progress of the treatment technology of VOCs produced by industrial emissions, and briefly introduces the performance characteristics of various treatment processes. The mainstream catalytic combustion treatment technology of VOCs is reviewed; the activity, stability and selectivity of the catalyst are discussed. This review has value for the researchers in the related areas. However, the paper needs improvement before acceptance for publication. My detailed comments are as follow:
1. In the manuscript authors should include the following relevant review articles related to catalysis:
a. doi.org/10.1007/s40089-021-00362-w
b. doi.org/10.3390/ijms222413383
2. Authors should include few references in the membrane separation and Photocatalytic oxidation section.
3. Authors should include a section of Prospects in future of this type of research.
4. If authors include few industrial facet that will be better.
5. The writing of summary and outlook section should be better.
6. There are few typos and grammatical errors.
Round 2
Reviewer 1 Report
The manuscript was improved, however, there are still a few details here that need to be addressed.
Please review the following abstract statement: " Industrial emissions of VOCs in the largest content of substances for benzene and hydrocarbons, benzene and hydrocarbons are also highly toxic substances." Benzene is a hydrocarbon. The review shows oxidation results for aromatics, polyaromatics (naphthalene), and short-chain hydrocarbons.
Please, complete the idea or add references for the following sentence: Line 216 “ Noble metal catalysts are mainly composed of noble metals such as Pd, Pt, and Au, Ag and their catalytic oxidation activity is easily affected by other factors, such as the addition of a second metal component.”
Please review the text on line 386.
Entire work needs revision/editing. There are missing spaces between words, there are citations with formats that do not conform to what is required by the journal, there are paragraphs with mixed font sizes, etc.
Some reviews of the subject have not been cited, for example a very recent one from Catalysts. Catalysts 2022, 12(10), 1134; https://doi.org/10.3390/catal12101134
Reviewer 2 Report
The authors took into account some suggestions for the revision of the article and some references were added at specific points. However, in my opinion, there are no substantial changes in the article that justify its publication in this journal. The references that were added are barely discussed as well as the rest of the analyzed papers and some important reports in the field are not cited. An example of that is the point about structured catalysts that was incorporated. Also, other important points only mention a few papers and the most important ones are missing. Therefore my recommendation is to reject this paper.
